# Uncovering Unique Concept Vectors through Latent Space Decomposition

**Mara Graziani**                                                           *mara.graziani@hevs.ch*
*Institute of Informatics*
*University of Applied Sciences of Western Switzerland (Hes-so Valais)*
*Sierre, CH 3960, Switzerland*

**Laura O' Mahony**                                                      *LauraA.OMahony@ul.ie*
*Department of Computer Science and Information Systems*
*University of Limerick*
*Limerick, V94 T9PX, Ireland*

**An-Phi Nguyen**                                                  *an-phi.nguyen@biognosys.com*
*Biognosys*
*Zurich, CH 8952, Switzerland*

**Henning Müller**                                                   *henning.mueller@hevs.ch*
*Institute of Informatics*
*University of Applied Sciences of Western Switzerland (Hes-so Valais)*
*Sierre, CH 3960, Switzerland*
*Department of Radiology and Medical Informatics*
*University of Geneva*
*Geneva, CH 1211, Switzerland*
*The Sense Innovation and Research Center*
*Sion, CH 1950, Switzerland*

**Vincent Andrearczyk**                                          *vincent.andrearczyk@hevs.ch*
*Institute of Informatics*
*University of Applied Sciences of Western Switzerland (Hes-so Valais)*
*Sierre, CH 3960, Switzerland*

**Reviewed on OpenReview:** *https://openreview.net/forum?id=LT4DXqUJTD*

## Abstract

Interpreting the inner workings of deep learning models is crucial for establishing trust and ensuring model safety. Concept-based explanations have emerged as a superior approach that is more interpretable than feature attribution estimates such as pixel saliency. However, defining the concepts for the interpretability analysis biases the explanations by the user's expectations on the concepts. To address this, we propose a novel post-hoc unsupervised method that automatically uncovers the concepts learned by deep models during training. By decomposing the latent space of a layer in singular vectors and refining them by unsupervised clustering, we uncover concept vectors aligned with directions of high variance that are relevant to the model prediction, and that point to semantically distinct concepts. Our extensive experiments reveal that the majority of our concepts are readily understandable to humans, exhibit coherency, and bear relevance to the task at hand. Moreover, we showcase the practical utility of our method in dataset exploration, where our concept vectors successfully identify outlier training samples affected by various confounding factors. This novel exploration technique has remarkable versatility to data types and model architectures and it will facilitate the identification of biases and the discovery of sources of error within training data.

# 1 Introduction

Understanding and explaining the inner workings of deep learning models is challenging, yet of outmost importance in high-risk applications where reliable and intuitive explanations are crucial for decision making. Model validation plays a vital role in ensuring trustworthy predictions and to avoid actions that may negatively impact society.

Because of their ease of use and understandability, the use of user-defined queries to inquire about the relevance of high-level concepts such as objects, shapes, and textures, has shown great promise in model validation. These concept-based queries are addressed through explainability methods that leverage concept vectors, representing vectors in the latent space of a layer that are representative of a concept (Kim et al., 2018). Methods further developing on concept vectors were proposed for multiple applications and tasks (Goyal et al., 2019; Graziani et al., 2020; Koh et al., 2020; Chen et al., 2020). However, concept vectors are generally obtained as a response to user-defined queries and are, as such, biased towards the user's knowledge and expectations, a phenomenon known as the experimenter bias (Rosenthal & Fode, 1963). Users may bias the analysis by only querying the model for some concepts while neglecting others. The exhaustive coverage of all possible concepts, besides, is unfeasible and in some domains such as biology or chemistry, the concepts may be unclear or difficult to define. Because of this, model interpretability with concept vectors is, as of now, limited to concepts that are well-identifiable and easy to label.

Achieving broader interpretability with concept vectors requires a reverse engineering approach that focuses on automating concept identification. In this case, the only assumption made on the concepts is that they are relevant for the models' downstream task. Few methods have ventured in the unsupervised exploration of imaging representations through activation maximization Erhan et al. (2009); Olah et al. (2017; 2020), clustering of the latents (Ghorbani et al., 2019; Klindt et al., 2023) or matrix factorization McGrath et al. (2022); Fel et al. (2023), but only limited analyses exist that match the discovered concepts with the downstream task activity (Fel et al., 2023). Considering the model's sensitivity to a concept, however, is essential to ensure that the concept discovery method focuses only on directions in the latent space that impact the model's output. In this work, we propose a we propose a novel framework that merges factorization and clustering of the latents together with output-sensitivity for concept discovery. As a result, we are able to isolate directions in the latent space that respond to well-distinguishable, unique concepts. Our novel approach can be applied with high versatility across multiple architectures, tasks, and data types.

To evaluate the effectiveness of our concept discovery approach, we apply it to the task of natural image classification. The discovered concepts align with human-understandable interpretations and demonstrate consistency with previous research findings. For instance, texture and object parts emerge as prominent in vision models, reflecting the work of Kim et al. (2018) on recognizing zebra stripes and Ghorbani et al. (2019) on detecting vehicle parts to classify police vans. Furthermore, user evaluation studies confirm that the discovered concepts are easy to understand, and show that they provide valuable insights about the model's decision making process.

# 2 Related Work

This work aims at explaining already trained models, focusing on post-hoc interpretability. Despite having acknowledged limitations (Rudin, 2019; Adebayo et al., 2018), post-hoc approaches allow us to gather important insights on de-facto standard models for vision tasks such as Inception (Szegedy et al., 2016) and ResNet (He et al., 2016), which are largely reused in real-world applications.

Post-hoc methods for vision models are numerous, and were categorized according to different taxonomies and criteria (Graziani et al., 2023). Four main groups can be identified covering (i) feature attribution, (ii) feature visualization, (iii) concept attribution and (iv) surrogate explanations. Feature and concept attribution methods, i.e. (i) and (iii) respectively, aim at identifying what characteristics in the data are the most related to the prediction. In the case of (i), this is done in the input space, with methods that evaluate the impact of perturbations by accessing the model gradients or estimating occlusion sensitivity. De-facto standard approaches are Class activation maps (CAM) (Zhou et al., 2016) and the gradient-weighted variant Grad-CAM by Selvaraju et al. (2017). Grad-CAM was actually demonstrated to be a generalization of CAM up to a normalization constant. Feature visualization (ii), initially proposed as activation maximization by Erhan et al. (2009) and further developed in the works Olah et al. (2017) and Olah et al. (2020), aims at uncovering the patterns that are learned at intermediate layers through visualizing what maximally activates each model unit. Finally, surrogate methods (iv) refer to techniques that require a surrogate model to generate explanations Ribeiro et al. (2016) The idea of concept attribution (iii) is particularly related to this work. Concept activation

vectors (CAVs) (Kim et al., 2018) were the first post-hoc method to introduce the hypothesis that linear combinations of units carry information on user-defined high-level concepts. Ghorbani et al. further extended the method to remove the need for user-defined queries (Ghorbani et al., 2019). The algorithm identifies image segments that cluster in the latent space, and uses CAVs to determine if the direction of a cluster is a relevant concept for the model. The identified vectors, however, depend on the quality of the image segments, which is not guaranteed for medical images (Graziani et al., 2021). Moreover, CAVs are unlikely to be orthogonal in the latent space. Counterintuitively, two CAVs for unrelated concepts may be closely aligned in the latent space. To alleviate this, Chen et al. (2020) proposed to train a concept whitening layer that ensures the orthogonality of the concept vectors in its latent space.

One could argue that research on concept discovery has started with mechanistic interpretability, and precisely with the interpretation of individual neuron activations through methods such as network dissection (Bau et al., 2017), activation maximization (Erhan et al., 2009) and feature visualization (Olah et al., 2017). These methods isolate one neuron in the model and consider it as a direction that spans the latent space. Different approaches were then proposed to obtain a visualization of what maximises the neuron's response. Research in this direction, however, has pointed out that neuron activations seem to respond to multiple unrelated features at the same time, a phenomenon known as *polysemanticity* (Nguyen et al., 2016; Olah et al., 2017; 2020) and that they do not seem to align to a single interpretable concept (Klindt et al., 2023), but rather to multiple ones at the same time (O'Mahony et al., 2023). Based on these observations, much of this research has shifted from considering individual neuron directions, towards considering combinations of multiple neurons at the same time (Bricken et al., 2023). If we consider the latent space as spanned by the neuron basis, such combinations of neurons can actually be interpreted as vectors pointing to specific areas of the latent space. Clustering methods were used in Ghorbani et al. (2019); Klindt et al. (2023); O'Mahony et al. (2023) to identify clusters of features and characterize them as concepts. Similarly, matrix factorization has been used by McGrath et al. (2022); Klindt et al. (2023); Fel et al. (2023) to obtain a decomposition of the latent space into a new basis that can be used for discovery purposes. The work by Fel et al. (2023), in particular, emphasises the importance of considering the network sensitivity alongside these directions, being in line with the observations in this paper. However, in contrast to this work, our method considers the additional use of clustering as a means of refining the concept directions after the sensitivity analysis, with the intent of isolating unique concepts, namely concepts with a clearly distinct meaning and a unique interpretation. K-means clustering was shown particularly beneficial to this end in recent works (Klindt et al., 2023; O'Mahony et al., 2023).

Not related to this work, but still relevant in the concept discovery realm are transversally different approaches that aim at using multimodality as a means of identifying concepts. For example, language models are used in Yan et al. (2023); Kalibhat et al. (2023) to identify coincise and descriptive attributes that are used in visual recognition, and that could therefore be considered as concepts. None of these methods, however, is highly versatile to different architectures and data types, and they mainly rely on additional multi-modal data such as images associated with textual captions and the use of multimodal foundation models such as CLIP (Radford et al., 2021). On a parallel line, an analysis of deep networks based on SVD was proposed by Raghu et al. (2017), where canonical correlation is used to test the similarity of two compressed representations of different model layers. Here the activations of a single neuron represent a vector in the canonical basis of the latent space. Following this line, non-negative matrix factorization was used to identify meaningful vectors to interpret AlphaZero (McGrath et al., 2022).

Our work takes inspiration from all of these works to introduce a novel method that does not rely on user-defined concepts or pixel segmentation, that does not require the training of additional concept whitening layers, and that leads to independent vectors pointing to unique concepts with a distinct semantic meaning.

## 3   Methods

Based on the observations of Kim et al., we assume that directions in the latent space of a layer can identify the concepts used by a model for inference (Kim et al., 2018). However, while they rely on user-defined concept examples to find the concept vectors, here we aim to reverse-engineer the problem. Our objective is to identify concept vectors that directly emerge as relevant directions impacting the model decision function. We further assume, as suggested by Chen et al. (2020), that concept vectors should be orthogonal to each other to maximize the separability of the concepts. the objective of our method is to identify the set of orthonormal vectors in the latent space that carry most of the information, i.e. that most impact the model's prediction.

The method consists of three phases:

1. variance alignment via the singular value decomposition (SVD) of the activations of a layer;

2. ranking of the singular vectors based on directional output sensitivity;

3. selection of the top ranked vectors and refinement of the direction to isolate *unique* concepts.

## 3.1 Notation

We consider the prediction function of a neural network $f : \mathbb{R}^m \to \mathbb{R}^n$ from an $m$-dimensional input vector to an $n$-dimensional output vector. We assume the model was already trained using a dataset consisting of $N$ labeled pairs of input data points and labels $\{(x_i, y_i)\}_{i=1}^N \subset \mathbb{R}^m \times \mathbb{R}^n$. Given an arbitrary layer $l$, the neural network can be seen as a composition of a feature extractor $\phi^l : \mathbb{R}^m \to \mathbb{R}^d$ and a downstream predictor $\psi^l : \mathbb{R}^d \to \mathbb{R}^n$, i.e. $f(x) = \psi^l(\phi^l(x))$. In the following, to simplify notation, and since we consider a single layer at a time, we drop the superscript identifying the layer $l$s. Furthermore, for a given input $x_i$, we use the shorthand $\phi_i = \phi(x_i)$. We are interested in finding $M$ orthonormal vectors $u_1, .., u_M \in \mathbb{R}^d$.

In the following, we present the method for $n = 1$ and for densely connected layers. The method can be extended to $n > 1$, for example multi-class classification by applying the same construction to each of the outputs. Similarly, for convolutional layers, a pooling operation is introduced to obtain a $d$-dimensional representation. Details about both extensions are given in Section 3.4.

## 3.2 Step 1: SVD

Step one aims at identifying the vectors that best summarize the encoding of the dataset in the latent space of a layer $l$ (with dimension $d$). To this end, we apply SVD to the entire layer's response to the input dataset. Such matrix $\Phi \in \mathbb{R}^{d \times N}$ is obtained by column-stacking the latent representations $\phi_i$, with $i = 1, \dots, N$. By applying SVD, we obtain:

$$\Phi = U \Sigma V^T \tag{1}$$

Note, $U \in \mathbb{R}^{d \times d}$, $V^T \in \mathbb{R}^{N \times N}$, and $\Sigma$ is a diagonal matrix of singular values $\sigma_1, \dots, \sigma_d$. The left singular vectors are the columns of the matrix $U$, and they align with the variance in $\Phi$. The singular values rank the directions from the largest to the lowest observed variance.

## 3.3 Step 2: Sensitivity-based ranking

At this point, we are only making use of the feature extractor $\phi$, and we are not considering whether the singular vectors are actually used by the downstream predictor. This means that the vectors found in the previous step might not be particularly relevant to the downstream predictive task. This second step evaluates the sensitivity of the output function along the directions of the singular vectors. The sensitivity represents the impact of perturbations of the feature representation of each input along the direction of the singular vector. As in Kim et al. (2018), this is computed as the derivative of the model output along the direction of the singular vectors. This operation is computed for all the singular vectors in $U$. Consider the gradient of the downstream predictor $\psi$ (Section 3.1) with respect to the latent space, which we denote $\nabla_\phi \psi_i$ for input $\phi_i$. To compute the directional derivatives, we rotate the gradients to align with the singular vectors in U: $\tilde{\nabla}_\phi \psi_i = U^T \nabla_\phi \psi_i$. At this point, it is important to note that an input data point may have large gradients for low-activation features, and vice-versa, high activation values may be annihilated by close-to-zero gradients. Therefore, we consider the joint impact of gradients and activations together. This operation is fundamental, as it reduces the gradient noise that can be derived by small gradient values at the end of model training. Let

$$\tilde{\phi}_i = U^T \phi_i \tag{2}$$

denote the coefficients of $\phi_i$ rotated to align with the singular vectors. We consider the (element-wise) product between the coefficients of the rotated activations $\tilde{\phi}_i$ and of the rotated gradients $\tilde{\nabla}_\phi \psi_i$:

$$\tilde{g}_i = \tilde{\nabla}_\phi \psi_i \odot \tilde{\phi}_i \tag{3}$$

Finally, to compute the ranking, we consider the overall importance of a singular vector to the prediction. This is simply given by the sample mean of the $g_i$ over all inputs $i$ in the dataset: $\tilde{g} = \frac{1}{N} \sum_i \tilde{g}_i$. Note, $\tilde{g} \in \mathbb{R}^d$. For simplicity

of the notation, we drop the tilde in the rest of the paper. Simply put, this step replaces the conventional ranking of the singular vectors given by the values in $\Sigma$ with a new ranking based on the value of the coefficients of the rotated gradients.

### 3.4 Extension to Vision Classification Tasks

Let us demonstrate how to extend Steps 1 and 2 to vision classification tasks. We consider a convolutional neural network (CNN) $f : \mathbb{R}^{h' \times w' \times c'} \to \mathbb{R}^K$ classifying an input image in $n = K$ classes. Particularly, $f_{i,k}$ is the predicted probability of input $x_i$ belonging to class $k$, namely $p(y = k | x = x_i)$. $N_k$ is the number of samples in class $k$, with $N = \sum_{k=1}^{K} N_k$. For convolutional layers, the feature extraction is $\phi^l : \mathbb{R}^{h' \times w' \times c} \to \mathbb{R}^{h \times w \times d}$ and it maps an input image $x_i$ to $d$ feature maps of width $w$ and height $h$.

To compute the SVD of Step 1, we aggregate the spatial information by a global average pooling operation, hence $\Phi \in \mathbb{R}^{d \times N}$, where $d$ is the number of channels. More precisely, by pooling, we reduce each $\phi(x_i) \in \mathbb{R}^{h \times w \times d}$ to a $d$-dimensional vector.

Step 2 is also modified. For each class $k$ in a $K$-classification task, we compute a separate $g_k \in \mathbb{R}^d$. To obtain the ranking, we further evaluate how each $g_k$ compares to the values obtained for the other classes. For instance, we compare $g_k$ to the distribution of the $g_k^-$ obtained for all the input data points that are not of class $k$, hence for all the $K$ classes except $k$.

Formally, for each class $k$ we compute the average of the projection of $g_k$ on the singular vectors:

$$g_k = \frac{1}{N_k} \sum_{i, y_i = k} U^T g_{i,k}, \tag{4}$$

where $g_{i,k} \in \mathbb{R}^d$ is the global average pooling of $\phi(x_i) \odot \nabla_\phi \psi_{i,k}$[1]. We then compute the sample mean and standard deviation of the values obtained for all the rest of the data, namely for all classes except $k$. The mean is obtained, for instance, by averaging over all the samples $x_i$ such that $y_i \neq k$:

$$g_k^- = \frac{1}{N - N_k} \sum_{i, y_i \neq k} U^T g_{i,k'}. \tag{5}$$

Similarly, the variance is computed as:

$$\sigma_k^{-2} = \frac{1}{N - N_k} \sum_{i, y_i \neq k} (U^T g_{i,k'} - g_{k'}^-)^2. \tag{6}$$

Finally, the importance scores for each singular vector in $U$ are given by:

$$z_k = \frac{g_k - g_k^-}{\sigma_k^-} \tag{7}$$

where $z_{k,j}$ is the importance score of the $j$-th column in $U$. This measure is then used to obtain a class-specific ranking, from which we retain the first $M$ positions to identify $M$ concept vectors.

### 3.5 Step 3: Identification of unique concept vectors

This step focuses on isolating directions that act as pointers to *unique* concept representations. Note, by *unique concept* we refer to a distinct and individual idea or notion that is well-distinguishable from others. Therefore, we aim at isolating a specific and well-defined characteristic or pattern that is relevant to the task at hand and semantically distinguishable from other identified concepts.

At this stage, the singular vectors found by SVD provide no guarantee on whether they are pointing to unique nor human-understandable concepts. To account for this, we propose an additional step, namely the disentanglement of

---

[1]Note, the pooling is here applied after the element-wise product of the features maps and the gradients

singular value directions into unique concept vectors that possess a distinct semantic interpretation. This refinement process employs an unsupervised clustering methodology that was previously investigated by O'Mahony et al. (2023) on individual neuron directions. The clustering requires six passages. We start by (i) aligning each latent representation $\phi_i$ in accordance with the directions indicated by the singular vectors. Then, (ii) we isolate the representations $\phi_i$ exhibiting the highest coefficients along each direction. This is done by looking at the magnitude of the components in each direction and retaining the inputs with the highest values. In (iii) we use hierarchical clustering to identify clusters of inputs with minimal intra-cluster distances. This step is used to identify the optimal number of clusters, which is determined based on a distance threshold parameter that controls the granularity of the concepts. At point (iv), if more than one cluster is identified as the optimal number of clusters, we perform an additional k-means clustering that further enhances the quality of the clusters. (v) Any outliers and insignificant clusters that contain an insufficient number of samples (i.e. fewer than five images) are removed . Finally, in (vi) we find the candidate concept vectors as the vectors pointing to the centroids of the newly identified clusters, namely by taking the mean activations of the respective clusters.

Human-interfacing is the last action of this step, necessary to interpret the discovered knowledge. The simplest way to obtain insights about a candidate vector is by data analysis and visualization. Depending on the data type and the model architecture, multiple approaches can be used to analyze the data. The simplest approach is to visualize the input data corresponding to the representations laying in the region pointed by the candidate concept vector. In dense models, the candidate vectors can also be used as feature importance estimates. The elements of a candidate vector $u$ are used as weights that are multiplied element-wise to the feature importance values obtained by back-propagating the model's gradients all the way to the input. Section 3.6 discusses more in detail how to obtain concept visualizations and interpretations in detail.

### 3.6 Concept Maps and Segmentation Masks

The main challenge of concept discovery is the visualization and interpretation of the discovered information itself. As McGrath et al. (2022) discuss, the discovered information may not always be interpretable nor associable with a human-understandable concept, and research is needed in this direction. A concept vector discovered by our method can be interpreted as a latent discriminant direction for the downstream task. The variance of the activations aligns with the concept vectors, and feature perturbations along these directions have the largest impact on the output function. Visualizing how a concept activates given a certain input can be particularly beneficial to the interpretation of the discovered information in human-understandable terms. We define concept activation maps as the visualization of the model's response, for a given concept, to a certain input. The computation of such maps is straightforward for convolutional networks. For a discovered concept direction $u \in \mathbb{R}^d$, and an input image $x_i$, we can visualize the layer's activation responding to the discovered concept as the sum of the $d$ feature maps for each channel $\phi_{i,1}, \ldots, \phi_{i,d}$ weighed by the concept vector coefficients. In other words, one can take the feature maps of each channel and weight them by the coefficient values of the concept vector[2]. Note, if we were to consider a concept that is fully aligned with the $n$-th feature map (with $n < d$, e.g.. $u = (0, ..., 0, 1, 0, ..., 0)$ (non-zero only for $u_n$), the visualization for $x_i$ would correspond to the $n$-th feature map itself. Concept segmentation masks can be directly derived by the concept maps as in Bau et al. (2017), namely by retaining the input pixel values with concept activation higher than the 80-th percentile in the concept maps.

### 3.7 Dataset Exploration and Outlier Detection

We showcase the utility of concept discovery for dataset exploration, demonstrating how the discovered concept vectors can be used to find input samples that are mislabeled or that contain confounding factors. The data is rotated to align in the space spanned by the singular vectors and anomalous samples are identified based on the statistical dispersion of the data For instance, we isolate 10% of the representations that fall outside of the interquartile range of the representation coefficients for the training data. The inputs corresponding to these representations are flagged for further inspection, as they are outliers that may contain artefacts or potentially misleading confounding factors.

---

[2]Being the concept vector a singular vector, it has norm of one.

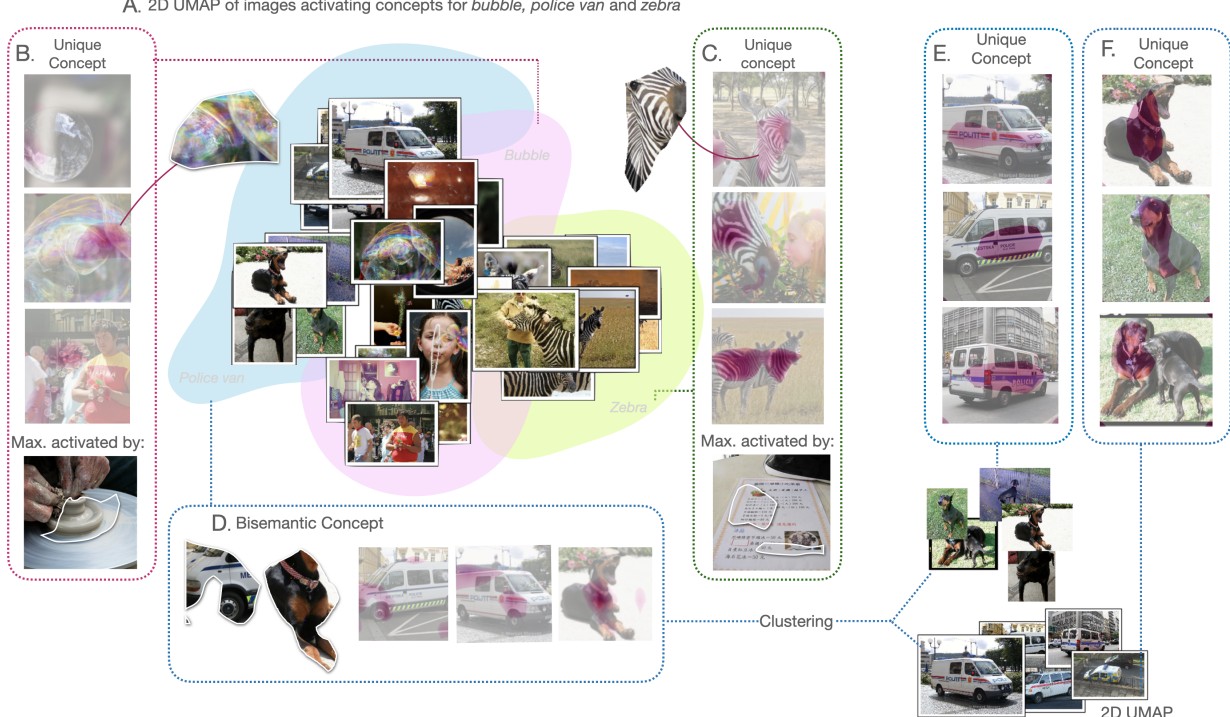

Figure 1: A. 2D UMAP of images activating the discovered concept vectors for *bubble*, *police van* and *zebra*. The images form three seperate clusters. We visualize concept segmentation masks for the images in each of the clusters. B. Concept segmentation masks and maximally activating input image for the class *bubble*. C. Concept segmentation masks for *zebra*. The concepts discovered in B. and C. are *unique* concepts. D. A bisemantic concept is identified for the class *police van*, activating for both vehicle parts and dog paws. After the refinement through clustering, the concept is split in two separate vectors representing either the graphics of the police logo or the dog fur. E. Unique concept representing the police log on the van structure. F. Unique concept deriving from the disentanglement of the polysemantic concept in D.

## 4 Results

This section presents the comprehensive evaluation of our concept discovery approach across various tasks and models. Our experiments focus on widely adopted models with publicly available pre-trained weights, which have been extensively studied in the field of interpretability. Specifically, we present the results obtained using Inception V3 (IV3) (Szegedy et al., 2016) with pretrained weights on the ImageNet ILSVRC2012 dataset (Russakovsky et al., 2015). In addition, we demonstrate the versatility of concept discovery beyond image-based applications by showcasing its effectiveness in classification tasks involving tabular data (details provided in the Supplementary Materials).

### 4.1 Controlled validation with ground truth

To validate our method's ability to accurately discover concepts, we perform controlled validation experiments. Following the approach in Kim et al. (2018), we define binary classification tasks where the crucial concept influencing the outcome was introduced by design. We selected the *church* and *tench* classes from ImageNet and build two binary classifiers with Inception V3 backbones. For the first classifier, denoted as IV3_noise0, the church training images are augmented by a red squared stamp, and the fish ones by a green stamp. The stamps were randomly placed within the images, with a noise level of zero, making them perfectly correlated with the class label. Our expectation was that the model would rely on the presence and color of these stamps for classification. When evaluating this model's performance on test images with and without the stamps, we could test this hypothesis. This analysis gives us, in fact, the ground-truth causal concept effect (CaCE) of the stamps, revealing the actual impact of the stamps on the

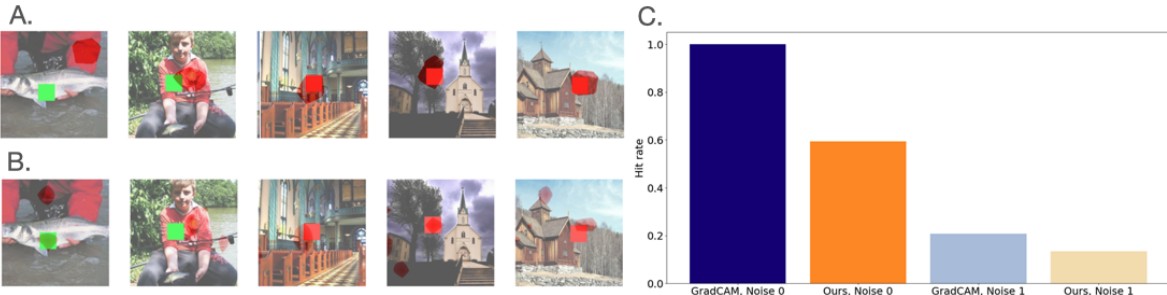

Figure 2: Visualization of IV3_noise0 heatmaps obtained by A. GradCAM and B. our concept activation maps method for the most important concept discovered by our method. The first two images are *tench* inputs misclassified as churches. Both methods point to the red jumper instead than to the green square. Similarly both methods point to the red squares in the correct images. C. Comparison of how frequently the two methods successfully hit the stamps.

model prediction (Goyal et al., 2019). After training IV3_noise0 from scratch for 100 epochs using stochastic gradient descent and standard parameters, the model achieved a remarkable test accuracy of 99.79% on images augmented with the squares. However, when the squares were removed from the test data, the accuracy plummeted to the level of random guessing (i.e., 51.73%), demonstrating a high CaCE. In contrast, the second model, named IV3_noise1, is trained on a dataset where the color assignment noise is set to one. Red and green squares were added randomly to the training images. In this case, the model reaches 95.43% test accuracy and the CaCE is low since the accuracy remains high at 93.34% even when the squares are removed from the test images.

We perform concept discovery in both models. The color of the red square is the most important concept discovered for IV3_noise0. As expected, this concept cannot be discovered in IV3_noise1. This shows that our discovery method only points to the stamps when these are actually used by the classifier. We quantify these observations in terms of the number of stamp hits over the entire dataset. Precisely, we count in how many images the heatmaps overlap with the stamp region and we then normalize this quantity by dividing it for the number of images in total. For IV3_noise0, our method correctly pinpoints the ground-truth concepts in the images with a frequency 83% at the Mixed 6e layer, and of 60 % at Mixed 7b. These values drop drastically for the concepts that we identify in IV3_noise1, for instance to 40% and 10%, respectively.

We compare both qualitatively and quantitatively our results to the object localizations obtained by GradCAM. A qualitative comparison is presented in Figures 2A and B. From the quantitative comparison of the hit rate in Figure 2C, GradCAM has higher true positive rates of the detection of the red squares, but it shows a higher rate of false positives than our method. This suggests that our approach has a reduced risk of misleading explanations for local explanations, e.g. for a single datapoint. Moreover, our method provides us with the additional information of the concept vector, thus allowing us to act on the concept itself by modifying, removing it or isolating it in the latent space.

## 4.2 Confirmation of existing findings in discovered concepts

Our method automatically identifies concepts associated with image categories. Our results are in line with specific findings that were already discussed by previous research (Kim et al., 2018; Ghorbani et al., 2019), specifically for the object categories of *zebra, police van and bubble*. Additional results for other classes than the selected ones and for ResNet50 are provided in the Supplementary Materials A and an integral analysis of the concepts discovered for ImageNet classes will be made accessible online.

We identify three orthogonal vectors as candidate concept vectors (i.e. one for each of the analyzed classes, with $M = 1$). The discovered concepts encompass a diverse range of visual elements, including patterns observed in the coat of *zebra*, graphical features and tires associated with a *police van*, and glossy reflections as seen in a *bubble*, as demonstrated in Figure 1. The maximally activating images are either visually similar or exaggerate the motifs shown by the activation maps. The maximally activating input for *bubble*, for example, exaggerates the glossy-like reflections and the round shapes generally observed in soap bubbles. The text of the menu restaurant contains black and white

stripes, which are also in *zebra* images. Importantly, these illustrations were obtained without any prior annotation of the concepts as detailed in Section 3.6.

Upon closer examination of the *police van* class (Figure1D, our analysis reveals an interesting observation. The candidate concept vector identified for this class exhibits responses to multiple unrelated features. In addition to activating for van tires and police graphics, it also responds to dog chests, which may seem unexpected at first. However, this phenomenon is not entirely unexpected and it can be attributed to the concept of *superposition*, which has been observed in CNNs (Olah et al., 2020). Superposition refers to the ability of neural networks to represent more features than the number of neurons they possess. This results in the presence of *polysemantic neurons*, which exhibit activation patterns that are semantically unrelated. Previous research highlighted the existence of such *polysemantic* neurons that activate simultaneously for car hoods and animal paws (Olah et al., 2020; O'Mahony et al., 2023), aligning with our findings. Our method demonstrates its capability to successfully disentangle the two activations associated with the *police van* class by identifying two distinct clusters that refine the concept direction into separate vectors, as illustrated in Figures 1E and F. In this process, the orthogonality of the direction is compromised to achieve unique concept directions for each activation. The disentanglement of the two patters allows us to capture the distinct semantic meanings associated with each activation, providing a more refined and comprehensive understanding of the underlying concepts.

### 4.3 Coherence and human interpretability of discovered concepts

To assess the coherence and human interpretability of the discovered concepts, we conducted a human evaluation test following the methodology outlined in previous works (Ghorbani et al., 2019). The test involved 30 participants, and the details of the evaluation test can be accessed at the following link: `https://forms.gle/MJ63G984ERvozuF38`. The evaluation comprised three main experiments, namely (i) intruder detection (ii) concept meaningfulness and (iii) inter-user agreement. Participants were familiarized with the test format by a prior introduction with an exemplar question and the corresponding correct answer.

In the intruder detection experiment (i) illustrated in Figure 3A, participants were presented with ten questions, each featuring a visualization of four concept segmentation masks associated with a specific concept vector, and one random mask from a different concept vector. The task was to identify the outlier image that conceptually differed from the others. Participants successfully recognized the intruders with average accuracy of $0.88$. Concept meaningfulness (ii) was measured by asking participants to label the concepts based on the visualization of the concept segmentation masks for three images. Despite the small image segments depicted the concept segmentation masks, all participants accurately labeled all the images, indicating that the concept segmentation masks were easily interpretable and could be associated with specific concepts. Inter-user agreement (iii) was evaluated by requesting participants to agree or disagree with concept labels provided by other participants. Out of the 30 users, inter-user agreement exceeded 91% for all questions. Notably, concepts identified for dog breeds showed relatively lower agreement scores, as users disagreed on the specific type of animal depicted in the concept segmentation masks.

### 4.4 Quantitative evaluation of concept relevance

We further measure the significance of the discovered concept vectors by quantifying the impact of concept removal using two different approaches, namely occlusion in the input space and weight annihilation at the intermediate layer.

Following the methodology of Ghorbani et al. (2019), occlusion is applied to the input pixels with high values in the concept map of each concept, specifically targeting the pixels with values above the 80th percentile. We quantify the smallest destroying concepts (SDC) as the smallest number of concepts to remove in order to see a performance degradation on at least $80\%$ of the dataset. Starting with the most important concept, we gradually remove concepts while monitoring the drop in performance. As depicted in Figure 3C, we observe a degradation of at least $80\%$ in the prediction accuracy when the first concept is removed for nearly $400$ classes. The performance degradation extends to almost all ImageNet classes ($962$ out of $1000$) when we remove the first five concepts. Supplementary Figure B.9 in the Supplementary Materials provides a comparison the model predictions before and after the removal of the SDC for each class.

One limitation of SDC is is the introduction of a distribution shift in the modified images due to the replacement of pixels in the input space. It becomes challenging to disentangle whether the prediction change is solely due to

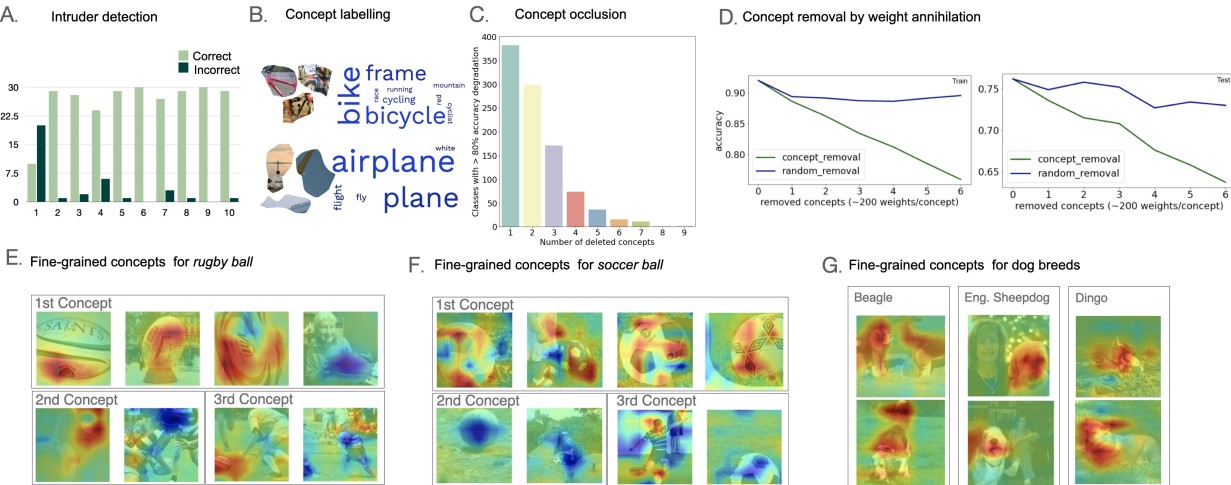

Figure 3: A. Detailed results of the intruder detection experiment with human participants. Out of ten classes, participants were able to detect with very high accuracy the intruding concept. B. Wordclouds of the concept labelling results for two concepts. The font size corresponds to the word frequency. C. Concept importance measured in terms of the impact of removing concepts in the input space by occlusion, namely by masking out areas with high values in the concept activation masks. Performance is evaluated on all ImageNet classes while gradually removing concepts starting from the most important. D. Concept importance measured as the impact of removing concepts in the latent space by annihilating the model weights that are linked to the concept. Performance is evaluated on all ImageNet classes while gradually removing concepts starting from the most important. E. Fine-grained concept activation maps for the first, second and third concept vectors differentiating rugby from soccer balls. The map adopts a symmetric colorbar with red for positive-, blue for negative- and green for zero-valued activations. Images are sorted from the largest absolute projection values on the concept vector. F. Fine-grained concept activation maps with symmetric colorbar for the concept activation maps on ImageWoof categories.

removing the concept or the distribution shift (Hooker et al., 2019). To address this concern, we conduct an additional analysis where concepts are dropped directly inside the layer, comparing the change in prediction accuracy against dropping random directions. To drop a concept, we retain the layer weights with low concept vector components and we set the remaining weights to zero. For example, to remove the first concept of the class *lionfish*, we set to zero $204$ parameters out of the original $2,048$ (keeping the remaining $1,844$ weight values untouched), based on the $80$-th percentile of the components of the concept vector for that class. We compare the impact of this ablation against dropping random weights to disentangle the effect of removing a specific concept direction from randomly setting weights to zero.

Figure 3D shows the accuracy drop observed when removing the concept directions from the first most important to the fifth, computed on the training images and 1000 validation images. Removing the first five concept directions causes an accuracy drop from $0.920$ to $0.785$ on train data, and from $0.762$ to $0.676$ on test data. The same drop cannot be observed instead when setting random weights to zero (i.e. accuracy $0.89$ on train and $0.734$ on test data).

## 4.5 Fine-grained analyzes

Concept discovery can be conducted in a fine-grained manner, focusing on specific subsets of classes. This targeted approach enables the identification of concepts that are tailored and relevant to a smaller group of similar classes. Such granular analyses are particularly valuable for classes that pose classification challenges or hold specific importance for interpretability analysis.

We consider a subset of ten dog breeds classes in ImageNet, referred as ImageWoof (Howard, 2019). This subset presents a significant challenge as the differences between the dog breeds are subtle (more details in the Supplementary Materials B). Our method identifies one concept for each class, totalling ten concepts. For *Dingo, Beagle and Old English sheepdog* we visualize the images that maximally activate the concept for each class, hence with the largest

projection on the concept vector (Figure 3G). High activation values are around the head for all dogs, and confusing factors are ignored (e.g. the woman in *sheepdog*). Each dog has a very distinctive head shape (the *dingo* has straight ears, the beagle long bent ears and the *sheepdog* has long black and white fur) and for this reason, different concept vectors are found for each class. The *beagle* also shows a diffused attention that includes the tail and paws of the dogs. This is probably to distinguish this class from the *English foxhound*.

By running concept discovery on pairs of classes, we identify concepts that provide informative insights into what distinguishes one class from another. For instance, Figures 3E and F showcase the concepts that differentiate a *Soccer ball* from a *Rugby ball*. In this analysis, we set the number of concepts to find for each class $M = 3$ and present three discovered concepts for each class in order of importance. The most important concept for *Rugby ball* emerges to be the shape of the ball, as shown by the activations around the ball in Figure 3E. On the other hand, the concept activation maps for *Soccer ball* identify the knitted pattern of the ball as the most important concept, and the soccer ball shape emerges only as the second most important concept. We can also see the attention to the context, particularly in the third concept. The maps activate on the players on the field, showing sensitivity to the player's jerseys and socks. Some of these findings align with the hypotheses found in (Ghorbani et al., 2019) about the relevance of jerseys to classify images in the *basketball* category. Thus, jerseys and socks may introduce confounding factors in the classification of these categories. Finally, in the inputs for the *rugby* class, the maps also activate on the players' hands holding the ball, or on the act of tackling, whereas in the *soccer* images the maps activate on the acts of kicking the ball.

## 4.6 Concept Uniqueness

We evaluate the uniqueness of the concepts identified by our method, showing that it is possible to disentangle superposition and identify vectors that point to unique concepts. Firstly, it is important to note that the discovered concepts do not align with individual model units. To quantify the alignment, we compute the cosine similarity between the discovered concept vectors for each category in ImageNet (considering $M = 1$ and a total of 100 concept vectors) and the basis vector of the latent space. The cosine similarity is consistently lower than 0.5 for various layers such as *Mixed_7b*, *Mixed_5b*, *Mixed_5d*, *Mixed_6c*, and *Mixed_7c*. A cosine similarity close to zero indicates the independence of the two directions, suggesting that the concept vectors and the latent space vectors are not aligned. In contrast, if the vectors were aligned, the absolute value of the cosine similarity would be close to one.

Next, we examine the outcomes of applying the disentanglement method proposed in O'Mahony et al. (2023). Specifically, we compare how many distinct clusters are suggested by the disentanglement method when applied to individual neurons as opposed to the singular vectors identified by singular value decomposition in our method. This analysis is informative on whether the basis of the singular vectors can facilitate capturing inherently distinct concept representations, i.e. *unique* concepts. Figure 4A, for example, illustrates how three separate clusters are identified as unique concepts starting from one of the singular vectors. The disentanglement outcome provides us with 3 distinct concepts for babies, teddy bears, and necklaces. By applying this approach to all of the singular vector directions within layer *Mixed_7b*, we quantify how many singular vector directions were already pointing to unique clusters and how many required disentanglement. As shown by Figures 4B and C, most of the singular vectors in our method already point to unique concept directions, and only a few require disentanglement. As opposed to individual neuron directions, fewer directions show polysemanticity. This means that the singular value decomposition step in our method is a necessary approach that facilitates the identification of unique concepts. The remaining singular vector directions that are not unique are then disentangled by our clustering method in distinct concepts.

## 4.7 Effective Outlier Detection

Our method successfully identifies 23% (equivalent 0.2% of the ImageNet training set) of the training images as outliers based on their representation in the latent space. Among these images, $184$ (1.8% of the training set) are misclassified by the model. Note, the flagged images were seen during training, hence they are challenging training examples that confuse the model rather than improving its robustness. This observation is supported by the lower top-1 accuracy on the flagged images at $0.90$ compared to $0.93$ on the rest of the training images.

Furthermore, when compared to random neurons or random directions, our method demonstrates higher accuracy in identifying these challenging training examples. The accuracy on the flagged images obtained through our method is $0.90$, whereas random neurons or vectors achieve an accuracy of $0.92$. This reaffirms the notion that the concept vectors discovered by our method offer more informative directions for dataset exploration than random alternatives.

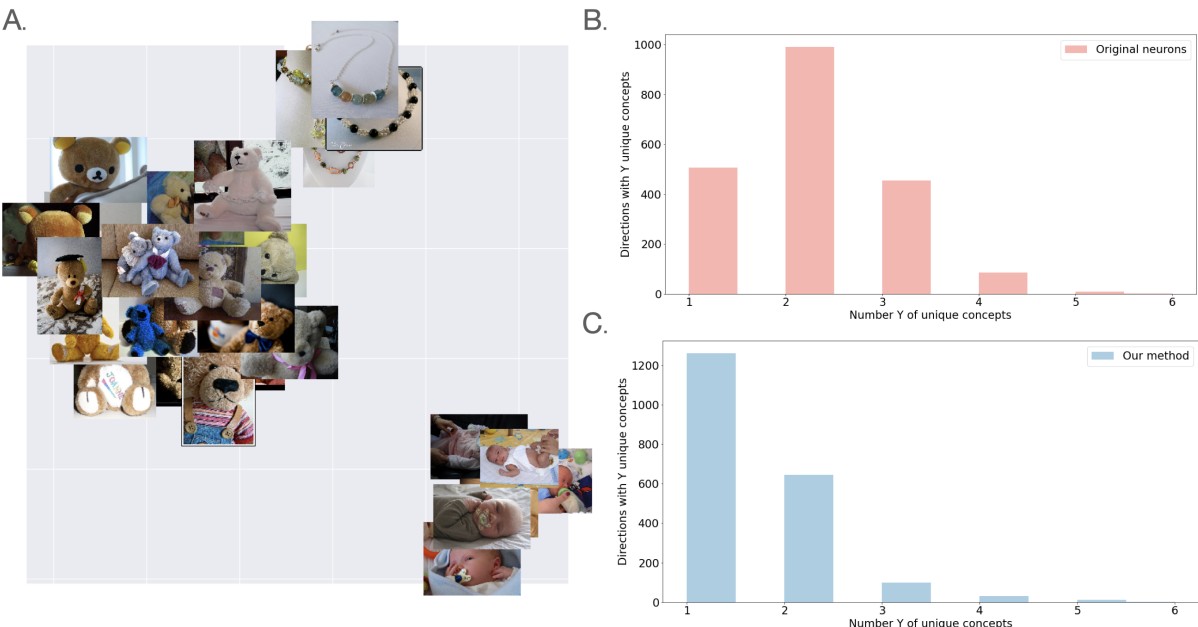

Figure 4: A. UMAP of the images maximally activating a polysemantic singular vector. Hierarchical clustering is used to identify the optimal number of clusters, i.e. three. Three concept vectors pointing to the centroids of each cluster are thus derived to obtain unique pointers to concepts. B. Number of clusters identified by hierarchical clustering for the images maximally activating individual neurons. The high number of neurons with at least two clusters shows that the directions are highly polysemantic, confirming existing research. C. Number of clusters identified by hierarchical clustering for the images that maximally activate our singular vectors. The high number of directions with a single cluster shows that the singular vectors are more likely to identify unisemantic concepts.

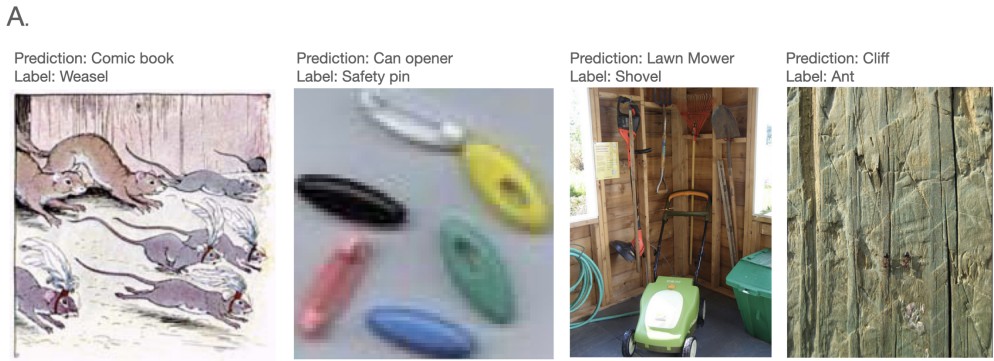

Figure 5: A. Results of dataset exploration. Our method identifies training images with distinct issues, such as style shift to drawings (1st image), poor resolution (2nd), multiple correct labels (3rd) and optical illusions (4th).

Figure 5 shows some examples of the outliers identified by our method. These images present notable variations in style and resolution (e.g., the first two images from the left) or contain confounding factors. For example, in the third image from the left, multiple labels (shovel and lawn mower) are equally correct. Additional examples are shown in the Supplementary Materials.

## 5 Discussion

The automatic discovery of the latent concepts used by complex models remains an under-explored research direction. Our approach offers a novel yet effective means to automatically discover concept vectors that point at image clusters with a unique semantic meaning. Our experiments demonstrated its potential and accuracy through diverse evaluations. We successfully identified the influential concepts in controlled experiments and we illustrated how removing the most important concepts impacts model performance.

A key strength of our concept discovery method is its independence from manual annotation, supervision and image preprocessing. This significantly reduces the time and effort required in traditional manual labeling methods, and it makes our method highly flexible and generalizable. Scientific domains such as biochemistry could strongly benefit from this novel approach, using it to uncover biases in heterogeneous datasets and provide insights into the molecules driving reaction prediction. In medical imaging, it could support knowledge discovery, such as identifying visual cues predictive of genetic alterations from biological tissue slides. The uniqueness of the discovered concepts is another noteworthy aspect of our approach. We demonstrate that our concept vectors can more appropriately capture independent and distinct visual attributes than individual neuron directions.

However, it is essential to acknowledge some limitations of this work. We did not compare our method to concept-based approaches like TCAV (Kim et al., 2018) and similar derivations (Graziani et al., 2018). This omission, however, is only due to the fact that these methods rely on sensitivity scores that our approach is designed to maximize in step 2. Thus, direct comparisons using these metrics would be biased in favor of our method. Additionally, concerns regarding the scalability of SVD on large datasets can be mitigated by our reliable results with only 1% of the dataset. Challenges may arise when applying our method to multi-modal models, where concepts may combine multiple modalities. Addressing this challenge would require back-propagating the information to individual modalities by saliency methods. Moreover, our methods's effectiveness relies on the quality of the input data used to compute SVD. If the network fails to capture relevant concepts during training, our method may not be able to discover them effectively. Finally, the evaluation of concept meaningfulness relies on human participants, which finally introduces some degree of subjectivity.

Nevertheless, our method proved effective in detecting input images with confounding issues, offering valuable insights on the models. The flagged outliers highlight images splotlight images with variations in style, resolution, or contain confounding factors. This information is valuable for improving model robustness and understanding the limitations and challenges associated with specific classes or image characteristics.

## 6 Acknowledgements

MG and VA devised the project, the main conceptual ideas and proof outline. MG worked out the technical details, carried out the implementation and experiments and wrote the manuscript with input from all authors. HM contributed to the revision of the paper and funding. LOM worked out the technical details and implementation of the clustering step, carried out experiments and contributed to the editing of the manuscript. AN contributed to the technical details and proof outline of the project. VA supervised the project, contributed to the technical details, planned the experiments, aided in extending the validation and editing the manuscript. All authors contributed to writing and editing.

MG acknowledges the support of the EU Horizon2020 project AI4Media (951911). LOM acknowledges the support of Science Foundation Ireland (18/CRT/6049). VA acknowledges the project TARGET (KFS-5549-02-2022-R).

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

# A  Additional Results on IV3 and ResNet50

Note: In our study, we apply the methodology to the 1000 classes in ImageNet. To ensure computational feasibility and accommodate our infrastructure capabilities, we employ an undersampling technique. Specifically, we retain only one out of every ten images from the ImageNet training dataset. This strategic choice enables us to manage memory requirements effectively and streamline the computational process. Nevertheless, we demonstrate how concepts can also be obtained by utilizing the complete set of available images for select classes, providing a more granular analysis in Section 4.5. Throughout the experiments, unless explicitly stated otherwise, we primarily focus on the concatenation layer named *Mixed 7b*. This layer is situated deep within the model, close to the end, and is expected to capture complex and high-level concepts. However, we emphasize that similar analyses can be conducted at different depths, layers, and even on entirely different architectures, as showcased in Appendix A.

We provide additional results for 16 classes, namely *airliner, clock, corkscrew, albatross, border collie, road sign, flamingo, mushroom, artichoke, hammerhead shark, screwdriver, iPod, tench fish, suspension bridge, umbrella* and cucumber. The concept segmentation masks for the first most important concept are shown in Figure E.13. The concepts were resized to fit in the square, but they originally appear at multiple scales in the input images.

Results obtained on ResNet50 (at *layer_4.0.add*) are shown in Figures E.14 and A.6. The visualizations point to concepts that are similar to those highlighted in IV3 visualizations, such as the fish fins, the zebra stripes, the glass-like reflections and van tires and logos. Figure A.6 shows the input images in the dataset that have the largest projection value on the concept vector for the four analysed classes. We can see the striped pattern emerging again for the class zebra, although in this case the color of the stripes seems to be given less importance. For the class *bubble*, the shape of the bird belly and their repeated presence may share a similarity with the images in the bubble class.

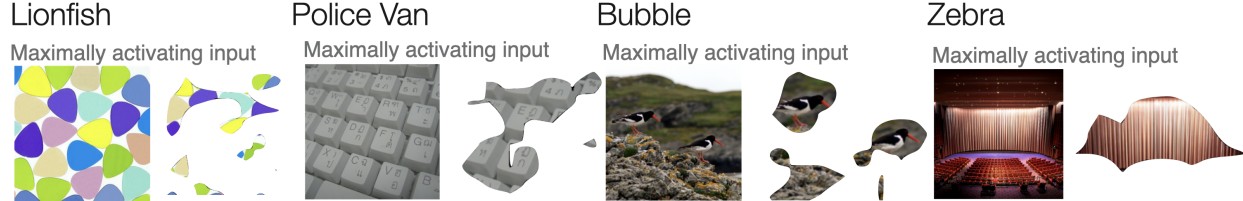

Figure A.6: Results on ResNet50. Visualization of the inputs with the largest projection value on the concept vectors for the classes *lionfish, police van, bubble and zebra*. Next to the input images we visualize the concept segmentation masks.

# B  Additional Results on ImageWoof

The ImageWoof dataset is a subset of the following ImageNet classes: *Australian terrier, Border terrier, Samoyed, Beagle, Shih-Tzu, English foxhound, Rhodesian ridgeback, Dingo, Golden retriever, Old English sheepdog*. An input image for each class is shown in Figure E.12. This dataset is challenging to classify because the classes represent multiple dog breeds with subtle differences. As Figure B.7 shows, the confusion matrix obtained on *training* images also presents misclassified samples.

Figure B.8 shows additional concept maps for the dog breeds ($M = 1$).

We show the prediction change before and after the removal of the smallest number of destroying concepts on Image-Woof in Figure B.9. No prediction change would appear as a diagonal line, whereas in our case we can clearly see that the prediction becomes random after removing the concept. The SDC is 1 for *Shih-Tzu, Rhodesian ridgeback, Beagle, Australian terrier and Golden retriever*, 2 for the *Old English sheepdog*, 3 for *Samoyed* and *Dingo* and 4 for the *Border Terrier*.

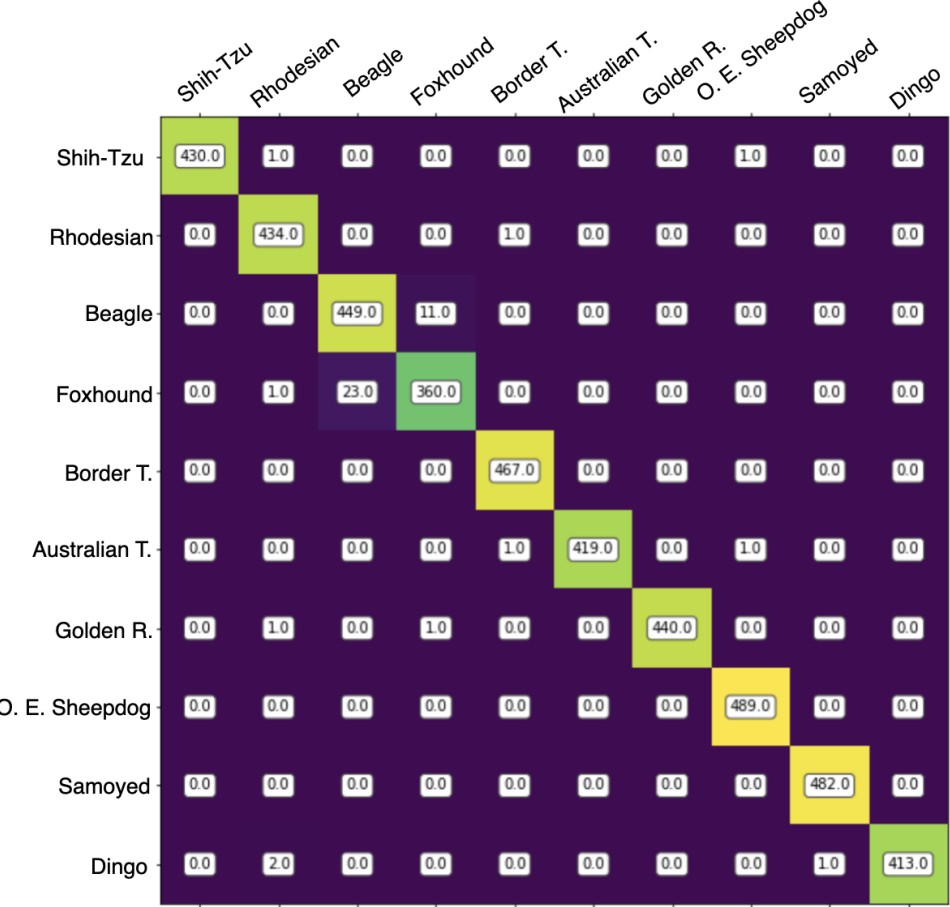

Figure B.7: Detail of the IV3 confusion matrix on ImageWoof classes.

## C  Additional Results from the User Evaluation

Figure C.10 reports in detail the performance of the end users on each question in the evaluation study part (i).

## D  Benchmark results on COMPAS

Here we demonstrate the applicability of concept discovery to non-imaging applications such as COMPAS. The model is, in this case, a densely connected feed-forward network. The COMPAS system uses criminal records and demographic features of nearly 19 thousand defendants to predict the likelihood of a new offense. Previous studies largely discussed whether race, gender and age of the defendants are sensible variables affecting the system's fairness (Jordan & Freiburger, 2015; Rudin et al., 2018). While we agree that a multilayer perceptron is unnecessarily complex for this task (Rudin, 2019), we recognize its importance as a contradictory task in the model interpretability literature.

Figure D.11 shows three discovered concepts for this task ($M = 3$). Note, gradient backpropagation is used to visualize the importance of each input feature for the concept vector. The values in the plot represent the importance given to the input features to project any input on the concept vector. The three concepts are a linear combination of multiple input features, and high relevance is given to the age of the defendant (age, in the first plot), the number of prior arrests (no_priors, in all plots), and the length of previous detentions (length_stay, two plots on the right).

The concept scores in the plot give a global explanation of the model. Local explanations about single inputs can be obtained by multiplying the scores with the feature values of each input. We benchmark the faithfulness of our explanations against other explainability methods by the Prediction Gap on Important feature perturbation (PGI) and

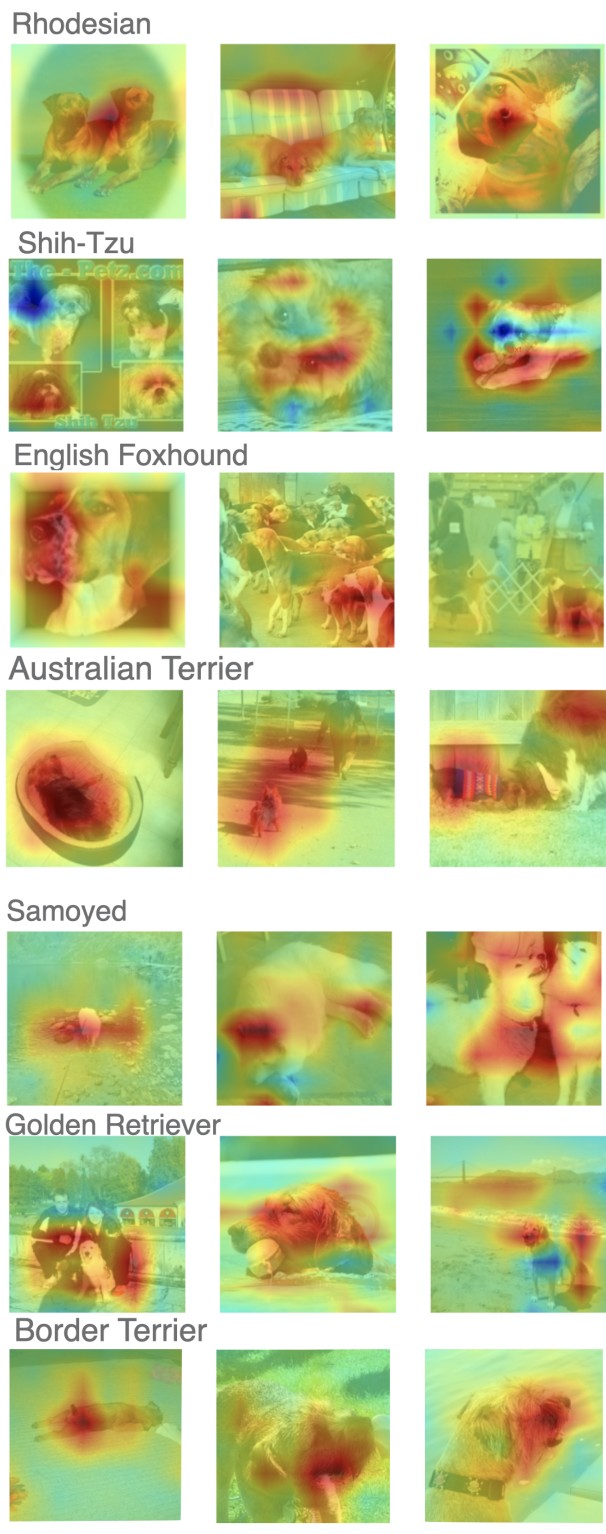

Figure B.8: Concept maps for the classes in ImageWoof. The images are ranked in order of their projection value on the concept vector (largest to lowest).

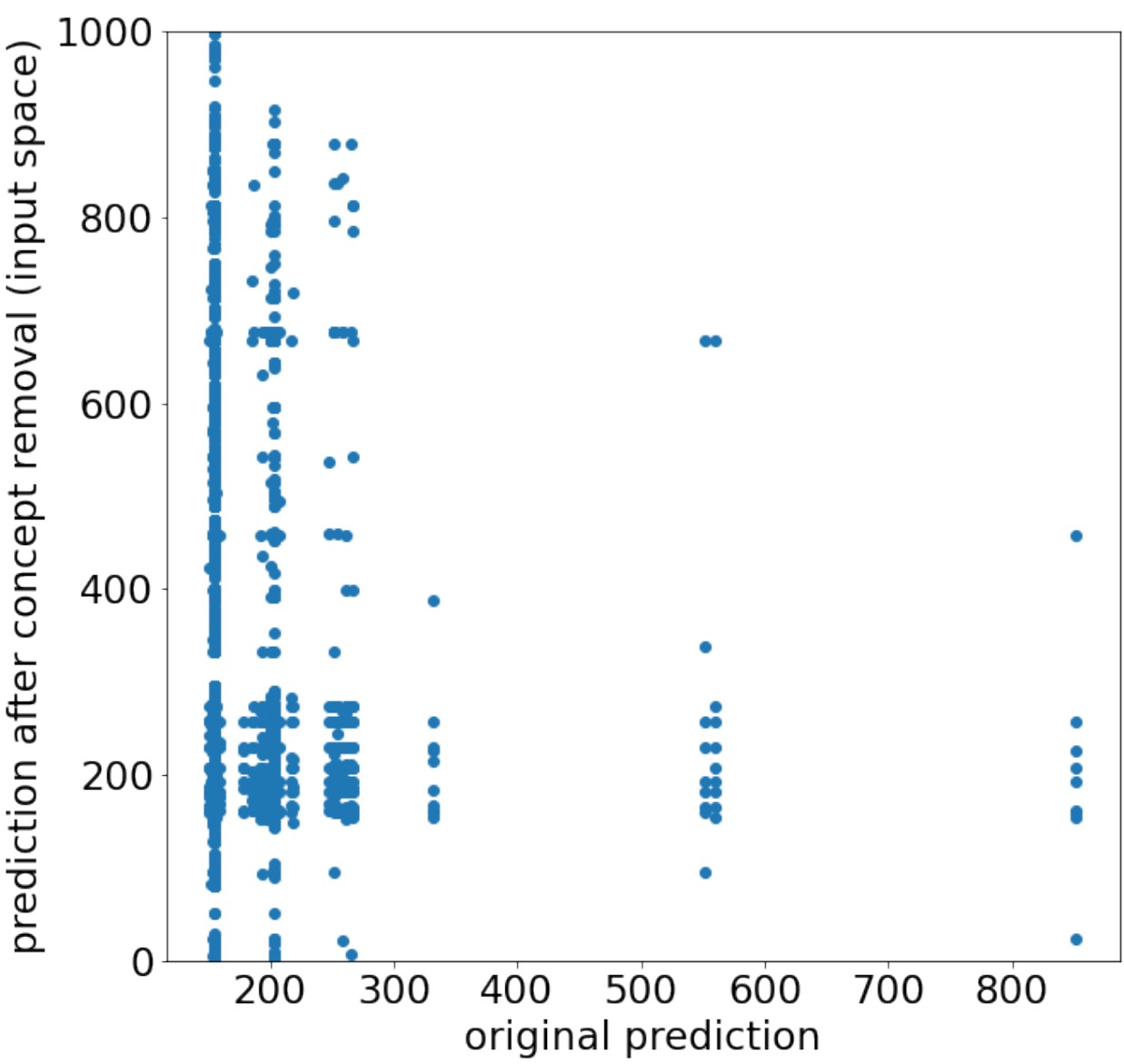

Figure B.9: IV3 predictions on ImageWoof before (original) and after concept removal in the input space.

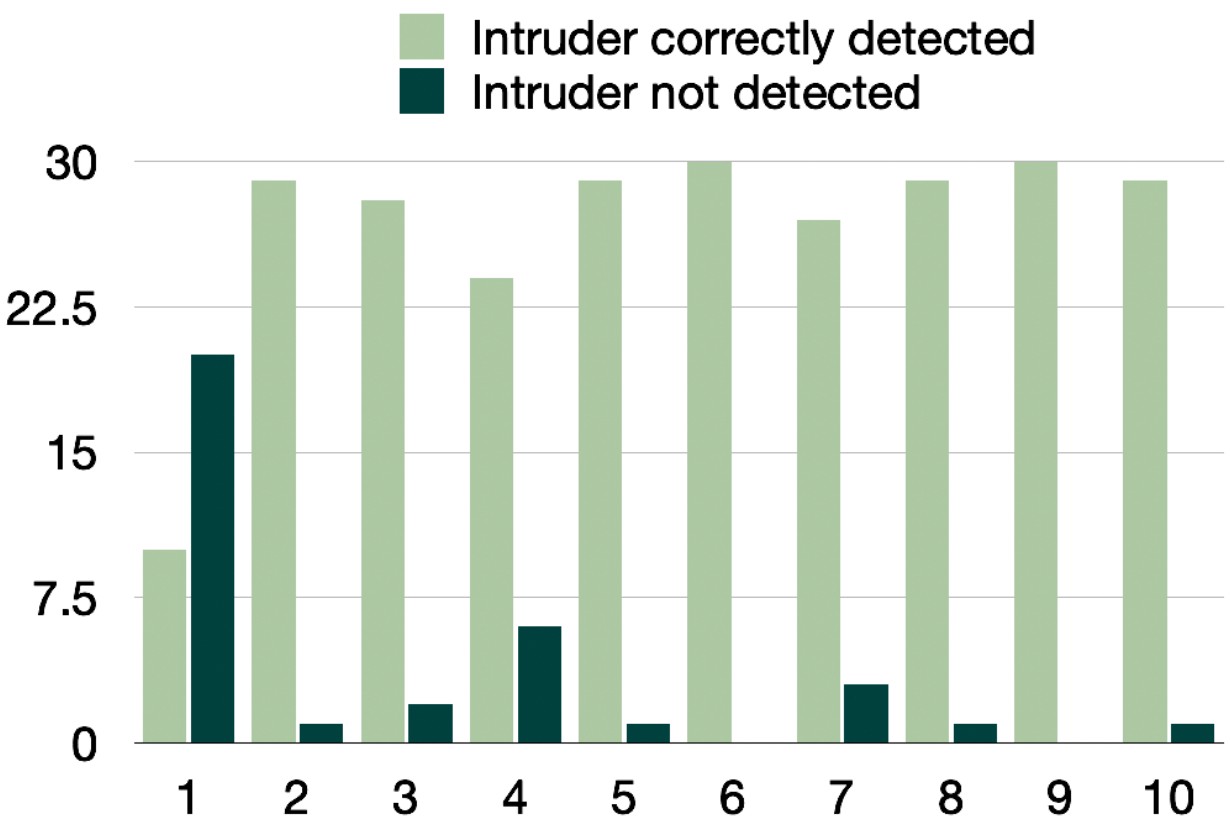

Figure C.10: Detailed performance on part i. of the human evaluation study.

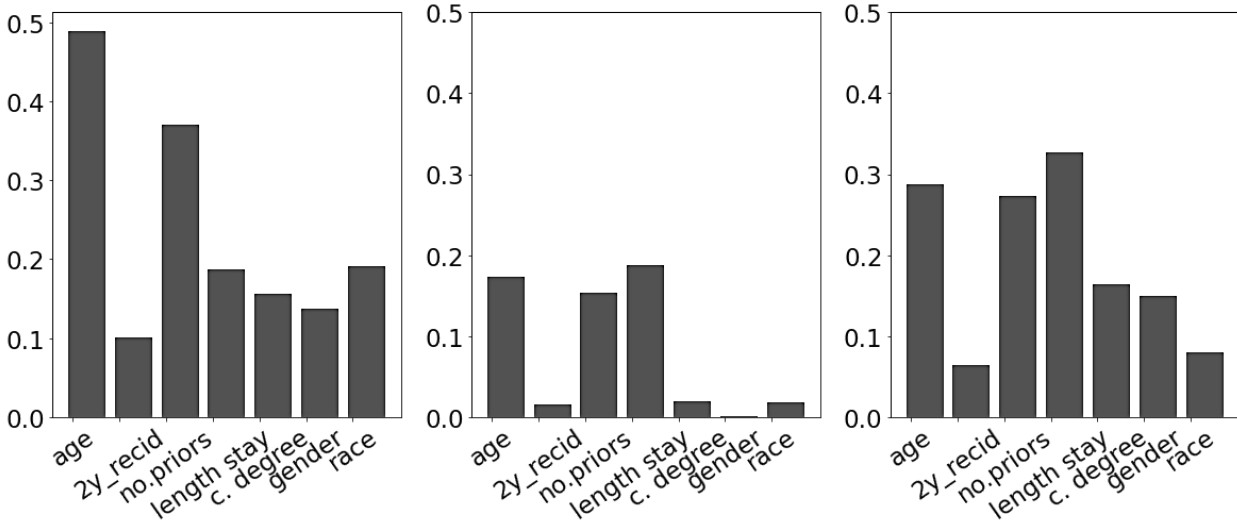

Figure D.11: Top 3 concepts discovered on COMPAS. The bar values illustrate the feature importance values to project each input on the discovered concept vector.

Unimportant feature perturbation (PGU), as in Agarwal et al. (2022). High PGI and low PGU values are desirable. The former indicates that the explanation identifies important features, while the latter that the method ignores unimportant ones. We compute the PGI and PGU following the benchmark code [3], obtaining 0.27 PGI and 0.0026 PGU. These values are comparable with other standard explainability methods, outperforming the benchmark results obtained by gradient-based approaches and LIME, as shown in the Appendix Table 1.

Table 1 benchmarks our method in the OpenXAI benchmark (Agarwal et al., 2022). The results in the table are taken, except for concept discovery, from the online leaderboard of the benchmark at `open-xai.github.io`. The hyperparameters of the explainer were set as illustrated in the benchmark instructions, i.e. {`protected_class:1;` `positive_outcome:1;perturbation_std:0.3`}.

Table 1: Explanation faithfulness for COMPAS.

| Method | PGI $\uparrow$ | PGU $\downarrow$ |
|---|---|---|
| LIME | 0.232 | 0.247 |
| Vanilla Gradient | 0.240 | 0.240 |
| Integrated Gradient | 0.240 | 0.238 |
| Gradient x Input | 0.254 | 0.216 |
| SHAP | 0.274 | 0.194 |
| SmoothGrad | **0.324** | 0.106 |
| **concept discovery** | 0.268 | **0.0026** |

# E  Additional Visualizations

---

[3]`github.com/AI4LIFE-GROUP/OpenXAI/blob/main/OpenXAI%20quickstart.ipynb`

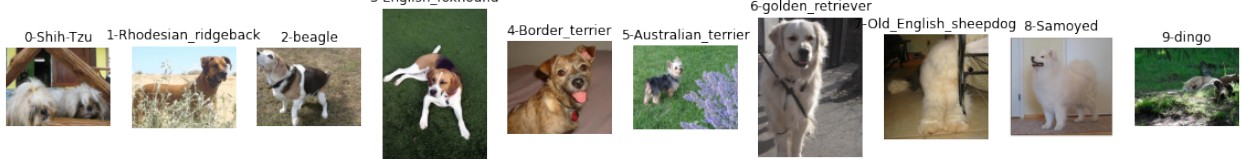

Figure E.12: Dog breeds in ImageWoof.

Figure E.13: IV3 concepts for additional classes. We show the resized segmentation masks.

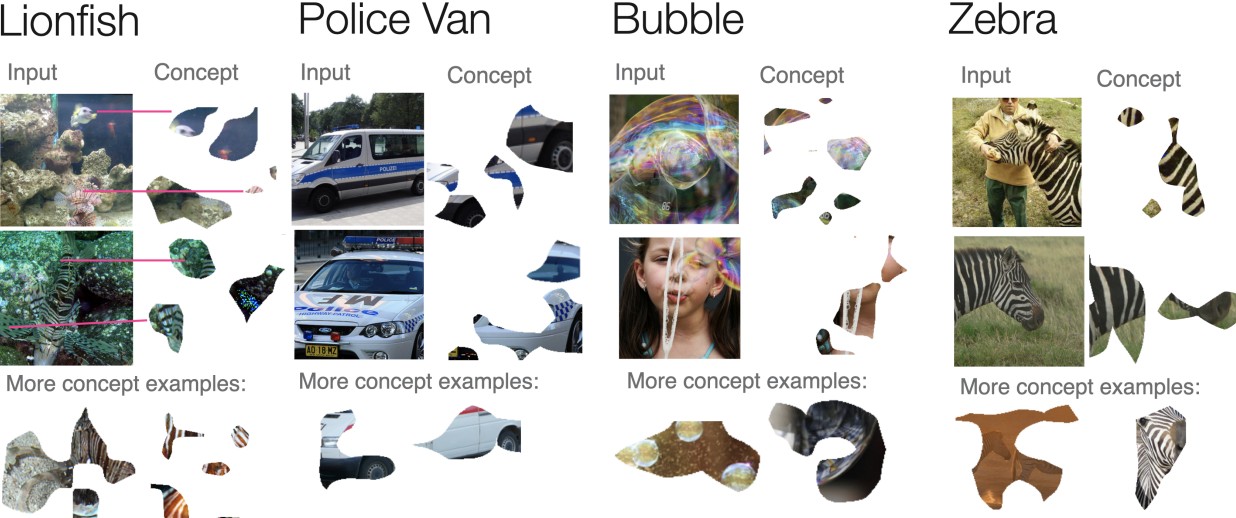

Figure E.14: Results on RestNet50. Segmentation masks of the concept vectors in *layer4.add* of ResNet50 for the same classes and input images in Figure 1.

