# OpenReview forum: "Uncovering Unique Concept Vectors through Latent Space Decomposition"
_TMLR — Accepted by TMLR_

### Review · Reviewer_Zkv5 · 2023-08-19

**Summary Of Contributions:**

This paper presents an approach to identify the key concepts, i.e., directions in latent space, that a model is relying on for its output. The approach consists of three steps: 1) perform SVD of the latent/representation space to identify the key directions of variance, 2) compute the sensitivity of the model's output along each important directions, and 3) decomposing the important SVD directions into concept directions via clustering. This approach is then used to analyze an Inception-v3 model (Resnet in appendix) trained on Imagenet. The paper also demonstrates several benefits of the proposal: e.g., outlier detection, and coherence.

**Audience:**

Yes

**Broader Impact Concerns:**

I don't see any ethical concerns with this work.

**Claims And Evidence:**

Yes

**Requested Changes:**

I'll describe the key broad changes here.

**Writing**:
See the weaknesses section. Here I believe the Section 3.5 needs to be rewritten to make all the key steps clear.

**Synthetic Example with Ground Truth**:
See the discussion from the weaknesses section. The key addition that I am seeking is a way to judge the effectiveness of the proposed method. I would like a setting where the key ground truth concept that a model relies on is known. The method here can then be tested against that setting to test whether this approach returns the key features that the model is relying on.

**Strengths And Weaknesses:**

### Strengths
- The concept discovery approach proposed here is simple to understand and quite straightforward. I particularly liked step 2 of this work, which measures the sensitivity of the model's output to the SVD directions.
- The paper demonstrates the effectiveness of the proposed approach across several kinds of tasks, e.g., outlier detection, showing that the output of the model is sensitive to changes in the concept, and showing that the concepts obtained are coherent, and unique.
- The paper also considers some challenging tasks such as fine-grained classification as was done for the dog breed setting.


### Weaknesses
- **Writing and Clarity**: the paper's writing could be much more improved. I'll raise some minor points later, but there are issues that detract from understanding the paper more carefully: 1) there are certain statements that are made without justification, e.g., "concept-based explanations have emerged as a superior approach that is more interpretable than feature attribution estimates such as pixel saliency". This statement is unjustified and likely doesn't belong in the abstract. 2) Typos(see last portion of the reviews). 3) Certain portions of the paper is quite dense to follow. For example, Section 3.5, which talks about the identification of unique concept vectors is quite unclear. This section should be rewritten to make the exact steps much more clear. For example, what is being clustered? Also, is section 3.6 part of the entire scheme? It seems like Section 3.6 is more a visualization step.

- **Evaluation & Synthetic Evaluation**: It is difficult to understand how reliable the approach presented here is. This is because the method is applied across a variety of settings and the results read like case studies. For example, the authors say that they confirm the previous research of Kim et. al. 2018 and Ghorbani et. al. 2019. However, one issue that plagues interpretability research and a lot of concept studies is the inherent confirmation bias of these procedures. Here is my suggestion for the authors. Can you design a toy setting where you train a model to rely on a specific concept apriori. You can then check to see whether you concept discovery approach is able to reliably recover that concept. One example of a testbed that might be useful here is the Spotcheck benchmark of Plumb et. al. from the paper "Towards a More Rigorous Science of Blindspot Discovery". In that test bed, they are able to manipulate the features that a model relies on. You can use a setting like that to demonstrate that this approach is effective. With a toy setting, you will be able to compute metrics that indicate how effective this approach should be compared to those of others. I believe this toy setting will be useful across the entire paper, for example, you can compute the effectiveness of the method for outlier detection, the uniqueness of the concept discovered compared to a ground truth. It is not enough to apply this method to realistic models, since we have no way to judge effectiveness in those settings; a synthetic controlled setting is required.

- **Related Work**: There is a large literature on using SVD/PCA on latent representations to discover important features that a model is relying on. This work should engage with that literature. For example, a closely related paper is ModelDiff: A Framework for Comparing Learning Algorithms by Shah et. al. 2023. It'll be important for this paper to clearly discuss these papers as well.

- **Details of the method**: There are a few degrees of freedom/hyper-parameters that need to be selected to use this method. For example, do you keep all the SVD directions, details of the clustering step etc. These details need to be added to the appendix.


**Some Typos, Writing Issues**
- Use citet and citep properly, e.g., Rosenthal & Fode (1963) in the introduction.
- Repeated 'we propose' in third to last line of the second paragraph of introduction.
- Discuss details of the evaluation test instead of linking to the form
- Section 4.3: 'is' repeated
- Section 4.5: what is M?

---

> ### Author Response · Authors · 2023-09-13
> **Response to reviewer's comments**
>
> Dear Reviewer,
>
> Thank you for your work! Your feedback is thoughtful and we are doing our best to address it.
>
> Starting from your most pressing concern, we have introduced new experiments in a setting where the key concept used by the model is known as ground truth, following the analysis in [1]. We trained two binary classifiers to discriminate images of fish from churches. For the first model, the church training images were augmented by a red colored box, and the fish ones by a green colored box. This gave us 100% correlation between the color of the box and the class label (i.e. the assignment noise is set to zero). For the second model, the color assignment was done randomly (i.e. the noise is 1). In the model with noise set to zero, the driving concept is the color of the red square, whereas this is not the case in the model with noise set to one. Note that, in the noise 0 model, we verify that the model is using the squares by evaluating the performance on images without squares and ensuring that the accuracy drops to that of random guessing.
>
> Our method discovers a concept vector that points to the red colored pixels as the most important concept for the church class. As expected, the “red square concept” is not relevant anymore in the model with noise set to one, as this is, in fact, not helping the model to perform the classification. We quantify these observations in terms of the number of hits over the dataset. Our method correctly pinpointed the ground-truth concepts in the images with a frequency 83% at the Mixed 6e layer, and of 60 % at Mixed 7b. These values drop drastically for the noise-1 model to 40% and 10%, respectively.
>
> As for your additional points, we are revising Section 3.5 to provide a more transparent and detailed account of the steps involved in the identification of unique concept vectors and we will clarify details about the hyper-parameters in our method. We are also carefully checking for all typos, reformulating unclear descriptions and missing literature references.
>
> [1] Kim, Been, et al. "Interpretability beyond feature attribution: Quantitative testing with concept activation vectors (tcav)." International conference on machine learning. PMLR, 2018.

---

### Review · Reviewer_HnoJ · 2023-08-20

**Summary Of Contributions:**

Understanding the inner workings of a deep learning models is crucial. One of the methods for understanding the results of deep learning models is concept vectors. Existing methodologies identified concept vectors through manual annotation or supervision. However, the authors propose a method that automatically identifies concept vectors using an unsupervised approach. A significant advantage of this method is its independence from the model architecture, task, and data type due to its automatic determination of concept vectors.
Moreover, the authors have empirically and quantitatively evaluated how well the deep learning model has learned the label information. Particularly, in the quantitative evaluation, they introduced a novel approach which involves reevaluating the performance of the model after retaining the weights of low-concept vector components in the intermediate layers while setting the weights of other significant components to zero. The effectiveness of the learned concept vectors becomes evident when observing maximal performance degradation.

**Audience:**

Yes

**Broader Impact Concerns:**

I do not see any concerns regarding ethical implications.

**Claims And Evidence:**

Yes

**Requested Changes:**

One area for improvement in this paper could be enhancing the objectivity of its results. The findings lack comparison against other pre-existing methods for deriving concept vectors. While I agree with the authors on the necessity of automatically deriving concept vectors using the proposed unsupervised method, this paper may benefit from further evidence to ascertain whether the proposed method outperforms other approaches. Therefore, it would be advantageous to incorporate comparative performance results alongside other methods.

**Strengths And Weaknesses:**

Strengths
-	The paper exhibited a well-organized overall structure, and including visual information and illustrative examples was beneficial for enhancing comprehension.
-	The method proposed in the paper for quantitatively evaluating the effectiveness of concept vectors is novel.
Weaknesses
-	The absence of a comparison with existing methods for deriving concept vectors makes it challenging to ascertain the advancements made by the proposed method.
-	The evaluation method for the learned concept vector, presented in section 4.2, is based on a small sample size of 30 participants, and the number of labels mentioned in each section is around 10, which is considerably limited. These few samples constrain an objective assessment on the efficacy of the method.
-	While the method describes the process of deriving the gradient through the element-wise product of the feature gradient and the feature, it lacks an explanation for adopting this approach over utilizing the feature gradient only.

---

> ### Author Response · Authors · 2023-09-13
>
> Dear Reviewer, thank you!
>
> Your main concern about the objectivity of the evaluation is a valid point, and we now extended our analyses.
> We performed new experiments following the benchmark in [1]. We developed two binary classifiers that differentiate between images of fish and images of churches. For the first model, the training images were augmented by a colored box that indicated the class label. The color was, in fact, perfectly correlated with the class label (i.e. the noise is set to 0). For the second model, the color assignment was done randomly (i.e. noise set to 1).
> When the correlation is maximal (i.e. noise is zero), the model focuses on the information in the colored square. We verify this by evaluating the performance on images without squares and ensuring that it drops to random chance. GradCAM [2] heat maps also illustrate high attention on the colored squares.
>
> Our approach successfully identifies the red colored box as a concept for the model trained on the noise-0 dataset. As expected, when the color assignment becomes random (i.e. the model is trained on the dataset with noise set to 1), the red square concept is no longer discriminative and thus not picked by our approach. A similar behavior is also observed in the GradCAM heatmaps.
> Our method correctly pinpointed the ground-truth concepts in the images with a frequency 83% at the Mixed 6e layer, and of 60 % at Mixed 7b. We believe that this analysis increases the objectivity of our evaluation, and shows that our method can reliably identify the underlying ground-truth concepts.
>
> We can now also compare both qualitatively and quantitatively to GradCAM. Our method does not outperform GradCAM in the true positive rates of the detection of the red squares, but it shows a higher rate of false positives. This is important as it shows that the risk of misleading explanations is reduced in local explanations, e.g. for a single datapoint. Moreover, our method provides us with the additional information of the concept vector, thus allowing us to act on the concept itself, for example by modifying, removing it or even simply isolating it in the latent space. These types of operations are only possible with concept vectors.
>
> Overall, we believe that our main contribution is a method that can discover concept vectors with great flexibility in the requirements for model architectures and data modalities. We focused on addressing this open need more than expressly trying to improve existing methods in terms of evaluation benchmarks.
> We have not neglected these benchmarks entirely, however. We do compare our method on a public benchmark for tabular data in Appendix D Table 1. Our results are comparable to, when not better than, other de-facto standard explainability approaches such as integrated gradients, smooth-grad, etc.
>
> As for a comparison to concept-based methods such as the TCAV [1] and similar derivations [3]: These methods are evaluated on the basis of sensitivity scores, hence the sensitivity of the model along the concept direction. Using these metrics as benchmarks would be unfair, since our method is built exactly to maximize this value by design. For this reason, we did not include a comparison to those methods and metrics, as our vectors would be favored in such evaluation.
> We are clarifying these points in the discussion and revising the paper accordingly.
>
> Finally,  we can add to the text an explanation of why we adopt the element-wise product of the feature and feature gradient instead of the feature gradient itself only. In short, the feature gradients tend to be noisy and often very small at the end of training. While this information is still useful and interesting, using only the gradients may be subject to noise and instability[4]. By considering the element-wise product of the feature and the gradients, we obtain more stable estimates that guarantee the focus on high-feature values and gradients at the same time. Moreover, by focusing on the element-wise product of features and gradients we ensure that our method only considers patterns that have high activation responses and that at the same time strongly impact the predictions.
>
> [1] Kim, Been, et al. "Interpretability beyond feature attribution: Quantitative testing with concept activation vectors (tcav)." International conference on machine learning. PMLR, 2018.
> [2] Selvaraju, Ramprasaath R. et al. “Grad-CAM: Visual Explanations from Deep Networks via Gradient-Based Localization.” International Journal of Computer Vision 128 (2016): 336-359.
> [3] Graziani, Mara, Vincent Andrearczyk, and Henning Müller. "Regression concept vectors for bidirectional explanations in histopathology." iMIMIC 2018, MICCAI 2018
> [4] Montavon, Grégoire, Wojciech Samek, and Klaus-Robert Müller. "Methods for interpreting and understanding deep neural networks." Digital signal processing 73 (2018): 1-15.

---

> > ### Comment · Reviewer_HnoJ · 2023-09-27
> >
> > Dear all,
> >
> > After reading the authors' responses, my concerns are now resolved. In particular, I agree that the authors quantitatively measured the proposed method's performance and explained why direct performance comparison with existing methods was not feasible. The authors proposed a novel method that works independently of the model architecture, task, and data type, and the proposed approach can work properly without human annotation. Additionally, I think that the previously identified weaknesses have now been resolved. In this respect, I would like to increase my rating about the paper from 'reject' to 'accept.'

---

### Review · Reviewer_SoMH · 2023-09-03

**Summary Of Contributions:**

The paper introduces a novel method for automatically discovering concepts learned by deep learning models during training. Through singular value decomposition and unsupervised clustering of the latent space of the model, the authors identify concept vectors aligned with directions of high variance that are semantically distinct and relevant to model predictions. The paper demonstrates the application of this method in natural image classification and its utility in dataset exploration and outlier detection. This work has the potential to enhance the interpretability of deep learning models and improve dataset quality.

**Audience:**

Yes

**Broader Impact Concerns:**

- Interpretability Challenge: The paper acknowledges that interpreting the discovered concept vectors, especially when they do not align with easily understandable human concepts, remains a challenge. This issue raises broader concerns about the practical applicability of the method in real-world scenarios where interpretability is crucial for decision-making.

- Computational Complexity: The method involves several steps, such as singular value decomposition and clustering, which may make it computationally expensive. This could limit its usability in resource-constrained environments, and addressing computational efficiency should be considered.

**Claims And Evidence:**

Yes

**Requested Changes:**

- Clarify Algorithm Steps: Provide a more detailed breakdown of each step of the proposed algorithm. Elaborate on the input, processes, and expected outcomes at each stage. This will assist readers in grasping the methodology with precision.
- Expand on Evaluation Metrics: If possible, extend the discussion of evaluation metrics used in the user studies. Explain the rationale behind the selection of specific metrics and how they relate to the method's performance. Include statistical analyses where applicable.
- Address Computational Efficiency: Given the potential computational complexity mentioned in the paper, discuss strategies or optimizations to address this issue. Explore how the method's efficiency can be improved for practical deployment.
- Generalization and Transferability: Emphasize the generalizability and transferability of the proposed method to different domains and applications. Discuss potential challenges or limitations when applying the approach to various contexts.

**Strengths And Weaknesses:**

**Strengths:**

Automatic Concept Discovery: The paper's method for automatically discovering concepts reduces the need for user-defined concepts, enhancing the method's versatility.

Wide Applicability: The approach shows promise across various architectures, tasks, and data types, increasing its practicality.

Broad Utility: The paper highlights the versatility of the method in dataset exploration and the identification of outlier samples, which can be valuable for quality control in machine learning applications.

**Weaknesses:**

Interpretability Challenge: While the method identifies concept vectors, the paper acknowledges the challenge of interpreting these vectors, especially when they may not align with easily understandable human concepts. Further research in this area is needed.

Complexity: The method involves several steps, including singular value decomposition and clustering, which may make it computationally expensive and less straightforward to implement.

---

> ### Author Response · Authors · 2023-09-18
> **Response and details on the algorithmic steps**
>
> Dear reviewer,
>
> Thanks for your work, your suggestions are useful and we can address most of your requests.
>
> To being with, we worked on a detailed summary of the method. For simplicity, we considered a vision regression model. Below, we provide the details on the inputs, outputs and the actions performed at each steps.
>
>
> Given i. a representative sample of the dataset, $X$; ii. a model to analyze, $f$; and iii. an intermediate layer for the analysis, $l$:
>
> 1. Perform a feed-forward pass on $X$ to extract the latent representations and the model gradients with respect to that layer. **Input:** $X$, $f$, $l$; **Outputs**: the latent representations at layer $l$, $\Phi$, and the model gradients with respect to layer $l$, $J$
>
> 2. Concatenate the latent representations of all inputs in $X$ in a single tensor. If the layer $l$ is a convolutional layer, then perform global average pooling on the spatial axes to obtain a matrix of the latents;
> **Input**: latent representations $\Phi$;
> **Output**: a matrix of the pooled latents, $\hat{\Phi}$
>
> 3. Compute SVD on the matrix of the latents $\hat{\Phi}$;
> **Input**: $\hat{\Phi}$;
> **Output**: the $U$, $S$, $V’$ matrices resulting from the SVD decomposition
>
> 4. Compute the sensitivity-based ranking of the singular vectors in $V’$.
> **Input**: the model gradients $J$, the latent representations $\Phi$, the singular vectors in $V’$;
> **Output**: an array with the ranking scores corresponding to each singular vector, $S’$
>
> 5. Analyze with unsupervised clustering the $M$-th singular vectors in the revised ranking to understand if they point to semantically unique features. Compute revised concept vectors if more than one cluster is found by pointing to the centroids
> **Input**: Cut-off threshold $M$ of the most important singular vectors to consider, the singular vectors in $V’$, a subsampling of the top input data points with maximal activation on the singular vectors, i.e. 100 data points suffice
> **Output**: unique concept vectors $\hat{V'}$, predicted cluster labels for each of the data points in the subsampling
>
> 6. Visualize the information corresponding to the most important singular vectors
> **Input**: $X$, $\Phi$, $M$, $\hat{V'}$
> **Output**: Concept activation maps for the top $M$ vectors in $\hat{V’}$ according to the $S’$ scores
>
> Motivated also by other reviewers, we have extended the evaluation of our methods by adding a comparison to Gradient-based Class Activation Maps (Grad-CAM) and we have introduced a controlled experiment with ground-truth concepts. We will add an in-depth discussion that discusses the advantages of evaluating the method in such controlled settings, and we will further extend our discussion on the previous evaluation experiments as well.
>
> In addition, we can comment on the computational costs. As this is a global post-hoc interpretability method, the computational time does not impact training nor inference. Besides, the understanding of the latents coming from the analysis with this method may be used for further reducing the time and computational complexity of training the models, for instance by removing some parameters that are used to learn unnecessary or unused concepts. As for the interpretability analysis per se, SVD can be expensive and not scale well on large dataset sizes. However, we have seen reliable results even just by using 1% of the dataset sizes. Our understanding is that a representative subsample of the training data is sufficient for performing the discovery analysis reliably, hence cutting considerably the computational costs. We are adding these points to the discussion in the manuscript.
>
> Finally, we will emphasize the discussion on the flexibility of our approach. In Appendix D, we apply our method to a multilayer perceptron for tabular data. Our explanations are at least as accurate as other standard methods in explainability.
> The scientific domain could strongly benefit from such flexibility. In bio-chemical tasks, heterogeneous datasets could be explored for biases due to underrepresented samples. Concept discovery could also give insights about the molecules driving reaction prediction. In medical imaging, it could be used for knowledge discovery, for example to discover visual cues that are predictive of genetic alterations from biological tissue slides by interpreting vision models such as the one in [1]. The main challenges we foresee concern the need for experts that can validate the discovered concepts. Moreover, the possibility to interpret models with multi-modal inputs is still to explore. Concepts would emerge as combinations of multiple modalities. Ideally, one could back-propagate the information to each individual modality by saliency methods to identify which modality contributes the most to define the concept vector.
>
> [1]Schmauch, Benoît, et al. "A deep learning model to predict RNA-Seq expression of tumours from whole slide images." Nature communications 11.1 (2020): 3877.

---

### Decision · Action_Editor_ib73 · 2023-11-10

**Recommendation:** Accept with minor revision

**Comment:**

This paper introduces a method for automatically discovering concepts learned by deep learning models. The approach identifies concept vectors in an unsupervised manner by performing SVD of the latent representation, ranking the singular vectors through sensitivity-based analysis, and decomposing the important SVD directions into concepts via clustering. Initially, the reviewers provided positive feedback on the submission, highlighting its generality and capability to handle any model architecture or task. They also noted its usefulness for dataset exploration and outlier detection. However, they raised concerns about the paper's clarity, the complexity of the approach, the lack of objectivity in evaluation, and the absence of comparison to explainability methods from the literature. During the discussion period, the authors addressed these concerns by modifying section 3.5 for improved clarity, conducting additional experiments for quantitative assessment, and providing comparisons to GradCam. They also explained why a direct comparison with concept methods might not be fair. Following these revisions and discussions, there was a consensus among reviewers to accept the paper.


The AE has carefully reviewed the submission and the discussions. The AE appreciates the rationale of the approach, which is sound and can bring important benefits in the field of concept explainability. The proposed unsupervised concept learning is convincingly validated by evidence. Particularly, the additional experiments and discussions provide much stronger validation of the proposed method. The applicability of the approach for dataset exploration and outlier detection is also highly valuable. While the discovered concept vectors may not be human understandable, the AE acknowledges this limitation, which can be addressed by future works in the field. The AE appreciates the improvements made in the paper's presentation and the inclusion of changes that were requested by the reviewers during the discussion period. Therefore, the AE recommends acceptance. As a minor change, the AE suggests that the authors expand the related work section to better position the paper's focus and put it into perspective with respect to state-of-the-art explainability approaches and the discussion in section 5.

**Audience:**

The submission addresses explainability, a significant challenge in machine learning that should capture the interest of a broad TMLR audience.

**Claims And Evidence:**

The claims regarding the concept explainability of the proposed method are clearly stated. The additional experiments conducted after discussions with the reviewers provide a strong validation of these claims.